# The Cell Membrane of a Novel *Rhizobium phaseoli* Strain Is the Crucial Target for Aluminium Toxicity and Tolerance

**DOI:** 10.3390/cells11050873

**Published:** 2022-03-03

**Authors:** Clabe Wekesa, John O. Muoma, Michael Reichelt, George O. Asudi, Alexandra C. U. Furch, Ralf Oelmüller

**Affiliations:** 1Department of Plant Physiology, Matthias Schleiden Institute of Genetics, Bioinformatics and Molecular Botany, Friedrich-Schiller-University Jena, Dornburger Str. 159, 07743 Jena, Germany; clabe.wekesa@uni-jena.de (C.W.); alexandra.furch@uni-jena.de (A.C.U.F.); 2Department of Biological Sciences, Masinde Muliro University of Science and Technology, P.O. Box 190-50100, Kakamega 50100, Kenya; jmuoma@mmust.ac.ke; 3Department of Biochemistry, Max Planck Institute for Chemical Ecology, Hans-Knöll-Str. 8, 07745 Jena, Germany; reichelt@ice.mpg.de; 4Department of Biochemistry, Microbiology and Biotechnology, Kenyatta University, P.O. Box 43844, Nairobi 00100, Kenya; asudi.george@ku.ac.ke

**Keywords:** aluminium toxicity, RNA-Seq, common bean, *Rhizobium phaseoli*, gene expression, aluminum tolerance

## Abstract

Soils with low pH and high aluminium (Al) contamination restrict common bean production, mainly due to adverse effects on rhizobia. We isolated a novel rhizobium strain, B3, from Kenyan soil which is more tolerant to Al stress than the widely used commercial strain CIAT899. B3 was resistant to 50 µM Al and recovered from 100 µM Al stress, while CIAT899 did not. Calcein labeling showed that less Al binds to the B3 membranes and less ATP and mScarlet-1 protein, a cytoplasmic marker, leaked out of B3 than CIAT899 cells in Al-containing media. Expression profiles showed that the primary targets of Al are genes involved in membrane biogenesis, metal ions binding and transport, carbohydrate, and amino acid metabolism and transport. The identified differentially expressed genes suggested that the intracellular γ-aminobutyric acid (GABA), glutathione (GSH), and amino acid levels, as well as the amount of the extracellular exopolysaccharide (EPS), might change during Al stress. Altered EPS levels could also influence biofilm formation. Therefore, these parameters were investigated in more detail. The GABA levels, extracellular EPS production, and biofilm formation increased, while GSH and amino acid level decreased. In conclusion, our comparative analysis identified genes that respond to Al stress in *R. phaseoli*. It appears that a large portion of the identified genes code for proteins stabilizing the plasma membrane. These genes might be helpful for future studies investigating the molecular basis of Al tolerance and the characterization of candidate rhizobial isolates that perform better in Al-contaminated soils than commercial strains.

## 1. Introduction

Common bean (*Phaseolus vulgaris*) is a major food worldwide. On average, the world consumption of common beans reached 2.51 kg per person in 2018 [1], an ~8% increase compared to 2008. As a legume, it forms symbiotic relationships with rhizobial bacteria, which enables the conversion of free atmospheric nitrogen (N) to ammonium, thus reducing the overutilization of N fertilizers in agriculture. Herridge et al. [2] demonstrated that the rhizobia fix between 50 and 70 million tonnes of N per year. They colonize the roots of the legumes and form nodules, which harbor oxygen-sensitive nitrogenase. Although the molecular basis of nodule formation and N fixation has been well investigated in the last decades [3,4,5], much less is known about abiotic stresses that inhibit symbiosis. In particular, besides salinity and drought stress, soils with low pH and therefore high aluminium (Al) contamination restrict common bean production worldwide, mainly due to severe impairment of nodule formation [6].

Al exists as insoluble salts, Al-silicates, and oxides [7] in soils. It is the third most abundant element (8%) in the earth’s crust after oxygen (47%) and silicon (28%) and the main reason for phytotoxicity in acid soils [8,9,10]. According to Kochian et al. [11], approximately half of the global arable land is acidic and exhibits Al toxicity to crops. Anthropogenic activities and, in particular, mineral leaching in the tropical soils have increased soil acidification up to the pHs between 5.5 and 4.6, and in extreme cases even below 4.5 [12]. Consequently, insoluble Al salts dissociate [13] in agricultural areas [14]. Plants counteract increasing Al stress in the soil by exuding organic acids and protons, immobilizing Al^3+^ by secretion of mucilage, sequestrating the ion in the vacuole, or secreting it from the root cells into the surrounding rhizosphere. These processes are often associated with activation of the antioxidant system in the host [9,12,15,16,17].

Al also affects the proliferation and growth of rhizobia in liquid media and soil [18]. Unlike in plants, Al toxicity is not only caused by its dissociation because of its acidity. Wood and Cooper [19] found that Al remained toxic to *Rhizobium trifoli* even after its precipitation from the media (at higher pH values). This implies that Al imparts harmful effects on soil bacteria over a wide pH range. Finally, it interferes with the nodulation process by inhibiting root hair formation and nodule initiation [20].

Al toxicity and tolerance mechanisms in rhizobia have not been conclusively elucidated. Johnson and Wood [21] proposed that Al^3+^ ions enter the cells of *R. trifoli*, bind to the DNA, and prevent its replication due to increased stability of the helical structure. Tolerance to Al by rhizobia was also correlated to its ability to withstand low pH [22]. A positive correlation was found between extracellular polysaccharide (EPS) production and rhizobia tolerance to acidy [23,24]; however, Tremaine and Miller (cited by [25]) ruled out the possibility that Al tolerance in rhizobia is due to EPSs complexion with Al ions. The authors compared different EPS-producing bacteria and showed that EPS production did not correlate with their Al tolerance. Additonally, Kingsley and Bohlool [25] showed that EPS was not responsible for Al tolerance in *Rhizobium leguminosarum*. The mutant strain with a lesion in EPS synthesis showed the same Al tolerance ability as the wild-type. 

Cytoplasmic resistance nodulation cell division (RND) pumps (also known as the CusCBA (CBA) efflux system) in the outer and inner membrane of the gram-negative bacteria have been proposed to participate in Al tolerance in rhizobia [6]. They associate with periplasmic adaptor proteins and outer membrane channels and export copper (Cu) and silver (Ag) ions to the external media [26]. Furthermore, the divalent metal efflux carrier DmeF confers metal tolerance in the bacterium *Cupriavidus metallidurans,* and a homolog of the *C. metallidurans’ dmeF* gene was found in *R. leguminosarum* [27]. It is highly expressed in response to cobalt (Co) and nickel (Ni) treatments, indicating that it suppresses toxicity caused by these ions. Consistent with this idea, a *dmeF* mutant showed an increased sensitivity to the two ions [27], but not to Zn (II), Cd (II), and Cu (II) ions. This may indicate that the tolerance to one metal is not uniform for all other heavy metal ions, including Al. Finally, since tolerance to Al toxicity correlated with the amounts of organic acids, such as citric acid and malic acids, in the culture media of rhizobia [13], these acids might participate in Al tolerance.

Previous studies have consistently observed that rhizobia tolerance to Al toxicity is strain-specific [13,25], perhaps due to their independent evolution under their respective environmental challenges. This variability calls for an independent investigation of rhizobia from a specific location when investigating their Al tolerance. For those studies, western Kenya soils are interesting and economically important because they possess pH as low as 4.9 and Al concentrations as high as 28 mM [28,29]. Commercially available rhizobia strains, such as the CIAT899, do not colonize common bean plants very well in Kenyan soils [28,30]. We previously isolated and characterized the novel rhizobial strains B3 and S2, which efficiently nodulated Kenyan common beans [31]. This study investigated how these rhizobial strains are adapted to the high Al concentrations in these soils. We included a novel isolate, called S3, from Nambale, Busia County (0°27′22.4” N 34015′52.8” E) and compared Al toxicity and tolerance of this strain with two recently characterized isolates from Kenya as well as with the commercial strain *R. tropici* CIAT899. Since B3 was the most tolerant strain to Al toxicity, it was selected for further analysis.

## 2. Materials and Methods

### 2.1. Rhizobia Exposure to Al Toxicity

Rhizobia were grown in a medium previously described by Wood and Cooper [19] at pH 5.5 with the following composition in µM: MgSO_4_·7H_2_O, 500; CaCl_2_·6H_2_O, 1000; KCl, 50; KH_2_PO_4_, 10; FeEDTA, 25; H_3_BO_3_, 10; ZnSO_4_·7H_2_O, 0.5; MnSO_4_·4H_2_O, 1; CuSO_4_·5H_2_O, 0.1; CoCl_2_·6H_2_O, 0.005; NaMoO_4_·2H_2_O, 0.025. After autoclaving, arabinose (0.3 g/L), galactose (0.3 g/L), sodium glutamate (1.8 g/L), biotin (250 µg/L), and thiamine (100 µg/L) were added as filter-sterilized solutions with a 0.2-micron syringe filter (Carl Roth, Karlsruhe, Germany). A total of 50 mL of the media was dispensed in 100 mL flasks, and a filter-sterilized solution of KAl(SO_4_)_2_·12H₂O was added to make 0, 25, 50, 100, 175, and 200 µM concentrations. All the reagents were purchased from Carl Roth (Carl Roth, Karlsruhe, Germany). An amount of 100 µL of bacteria cells at OD_600_ = 0.5, initially cultured in yeast extract mannitol broth (YMB), was added to each flask and incubated in the incubator shaker (150 rpm) at 30 °C. After 48 h, cell density was measured at OD_600_. The experiment was independently repeated four times. Growth inhibition by each Al concentration was calculated by the equation:$$\%\;\mathrm{inhibition}=\left(1-\frac{{\mathrm{Abs}}_{\mathrm{Treatment}}}{{\mathrm{Abs}}_{\mathrm{Control}}}\right)\times 100$$
Abs_Treatment_: absorbance at 25, 50, 125, and 200 µM KAl(SO_4_)_2_·12H₂OAbs_Control_: absorbance at 0 µM of KAl(SO_4_)_2_·12H₂O

### 2.2. Cell Viability after a Short-Time Al Exposure and Recovery after Longtime Exposure

Cell viability was performed by counting colony-forming units (CFUs) on yeast extract mannitol agar (YMA) plates. Cells of B3 and CIAT899 cultured in 50 mL YMB for 24 h were harvested by centrifugation for 10 min at 3300× *g* and washed two times in saline solution (0.85% NaCl). They were then transferred to a 100 mL conical flask containing 50 mL of defined glutamate-arabinose-galactose-salt media [19]. The two strains were treated with 0, 25, and 100 µM of Al^3+^ ions. After 12 h, 1 mL of the sample was retrieved, serially diluted 10-fold from 10^−1^ to 10^−10,^ and 100 µL was retrieved on every diluent to perform plate count. The experiment was independently repeated three times. For long exposure, isolate B3 and CIAT899 cells were exposed to 0, 25, 50, 100, and 200 µM of Al^3+^ ions in glutamate-arabinose-galactose-salt media an incubator shaker at 30 °C for 96 h. One mL was then retrieved and harvested by centrifugation at 5000× *g* for 5 min at 4 °C. They were then washed three times in sterile saline water. These Al-free cells were then used to inoculate 50 mL of freshly prepared YMB. They were incubated in an orbital shaker maintained at 30 °C. After 48 h, absorbance at 600 nm was measured in an Eppendorf BioSpectrometer^®^ basic (Hamburg, Germany). The experiment was repeated four times. The CFUs per mL counts were transformed to log_10,_ and together with OD_600_ values from the longtime exposure study, they were visualized with the seaborn package [32] in python 3.10. One-way ANOVA was then performed to compare the mean difference between untreated (0 Al^3+^ treatment) and treated cells in both viability and longtime recovery studies with Scipy v1.0 [33], followed by Tukey’s HSD (honestly significant difference) for pairwise mean comparisons with statsmodels v0.13.1 [34] in python 3.10.

### 2.3. Al Localization to the Cell Membrane

To investigate whether the cell membrane is one of the possible Al toxicity sites in rhizobia, we performed calcein (bis[N,N]-bis(carboxymethyl)aminomethyl) fluorescein staining [35,36]. Calcein (Thermo Fisher Scientific, Waltham, MA, USA) reacts with Al in an acidic environment (pH 5–6) to form a fluorescence 2:3 Calcein-aluminum compound. The cells previously cultured for 24 h in defined glutamate media were exposed to 50 µM or 100 µM Al overnight on a 150 rpm orbital shaker at 30 °C. They were then harvested by centrifugation for 10 min at 3300× *g* at 4 °C and washed four times to remove any unbound Al in 0.5 M MES (2-(N-morpholino) ethanesulfonic acid) buffer (pH 5.5). The cell pellet was finally resuspended in 5 mL of the MES buffer. In total, 5 µL of 20 mM calcein (Thermo Fisher Scientific, Waltham, MA, USA) stock (prepared in dry dimethyl sulfoxide freshly before the start of the experiment and stored on ice in the dark) was added to the samples. Cells were put in an incubator (New Brunswick Innova™^®^, Edison, NJ, United States) at 30 °C and shaken at 100 rpm. After 2 h, they were washed three times with the MES buffer to remove any unbound calcein, followed by gentle centrifugation at 3000× *g* for 10 min. Fluorescence was then recorded using a microplate reader (BMG Labtech, Ortenberg, Germany), where excitation and emission were set at 510 nm and 490 nm, respectively. The fluorescence intensity emitted is proportional to the membrane-bound Al quantity [36]. One-way ANOVA was performed to determine the mean difference in fluorescence intensity between isolate B3 and CIAT899. 

### 2.4. Membrane Damage

The leakage of ATP and fluorescence protein mScarlet-1 into the media was used in assessing membrane damage after exposing cells to Al. Overnight cultures were washed three times in MES buffer (pH 5.5) with centrifugation at 5000× *g* for 10 min. They were then suspended in MES buffer, treated with 50 µM Al^3+^ and incubated at 30 °C in an orbital shaker. After 4 h, they were centrifuged at 12,000× *g* for 10 min, and the ATP in the supernatant was determined using BacTiter-Glo™ Microbial Cell Viability Assay kit (Promega, Madison, WI, USA). This assay is based on the ATP’s ability to react with luciferin to form a luminescence complex. The Luminoskan™ Microplate Luminometer (Thermo Fisher Scientific, Waltham, MA, USA) measured the resulting luminescence. Luminescence values were normalized by calculating *L*/*L*_max_ (luminescence counts per s/total luminescence counts remaining).

mScerlet-1 is an engineered fluorescence red protein [37] that was constitutively expressed in *E. coli* when cloned in the plasmid [38]. To determine its leakage in the media, we transformed isolate B3 with the plasmid pMRE165 following the method described by Garg et al. [39]. Plasmid DNA was extracted from *E. coli* cells and isolated with GeneJET plasmid miniprep kit (Thermo Fisher Scientific, Waltham, MA, USA) following the manufacturer’s protocol. In an incubator shaker, a loopful of isolate B3 suspension was inoculated in 50 mL YMB and maintained at 30 °C for 48 h. Cells were chilled on ice for 30 min and then harvested by centrifugation at 3300× *g* at 4 °C for 10 min. The cell pellet was washed four times in cold, sterile deionized distilled water and finally in 10% glycerol. The cells were then resuspended in 5 mL 10% glycerol and kept on ice. A total of 2 µg of plasmid DNA was put in 90 µL of the cell suspension and mixed by vortexing at high speed for 10 s before incubation on ice for 30 min. The plasmid-cell mixture was loaded on a chilled electroporation plate and exposed to a single dosage of high voltage in a BTX’s ECM600 Electrocell Manipulator (South San Francisco, CA, USA) that generated a field strength of 25 kV/cm with a 0.1 cm gap cuvette. The cuvette was kept on ice for 10 min and then spread on YMA plates supplemented with 50 µg of kanamycin and 10 µg of tetracycline. The plates were then incubated for up to 7 days in an incubator at 30 °C. Colonies emitting fluorescence after observation in Axio Imager.M2 fluorescence microscope (Zeiss Microscopy GmbH, Jena, Germany) were scored as positive for mScarlet-1. For the leakage assay, 1 mL of mScarlet-1 positive cells were centrifuged for 5 min at 4 °C at 5000× *g*. The media was removed, and the pellet was washed three times with MES buffer (pH 5.5). The cells were resuspended in 1 mL MES buffer and treated with 0 µM, 25 µM, and 100 µM of Al^3+^ ions and incubated at 30 °C. After 1 h, they were centrifuged at 10,000× *g* for 5 min. The fluorescence from the supernatant was measured with the microplate reader; the excitation wavelength was 569 nm, and the emission wavelength was 594 nm. The experiment was repeated five times. 

Luminescence and fluorescence intensity was reported as fold change of Al-treated against the untreated cells. One-way ANOVA was performed to determine the significant difference of the leaked ATP between CIAT899 and B3 and mScarlet-1 leakage between treated and untreated cells. 

### 2.5. Mechanisms of Tolerance to Al Stress

We implemented global gene expression in *R. phaseoli* strain B3 to elucidate tolerance mechanisms to Al toxicity in rhizobia. This strain was chosen for this experiment because of its high tolerance to Al-contaminated media.

#### 2.5.1. RNA Isolation and Sequencing

Cells pre-cultured in YMB were used to inoculate the glutamate-arabinose-galactose-salt media at pH 5.0 either treated with 100 µM Al^3+^ in the form of KAl(SO₄)₂·12H₂O or untreated control and with three replications for each treatment. Cultures were incubated in a 150 rpm orbital shaker at 30 °C. After 48 h, 1.5 mL cells at OD_600_ = 0.5 were harvested by centrifugation at 5000× *g* at 4 °C for 10 min. RNA was then extracted with TRIzol^®^ reagent (Invitrogen, Waltham, MA, USA) following the manufacturer’s recommendation. Briefly, cells were immediately homogenized in TRIzol^®^ reagent and incubated at room temperature for 10 min. An amount of 200 µL of chloroform was added, mixed, and chilled on ice for 3 min, then centrifuged at 12,000× *g* for 15 min at 4 °C. The upper aqueous layer was carefully transferred to a 1.5 mL tube containing 500 µL of the isopropanol and incubated at room temperature for 10 min. The samples were then centrifuged at 12,000× *g* for 10 min at 4 °C. Isopropanol was removed carefully, and the pellet was washed twice with 1 mL of 75% ethanol. The sample was centrifuged at 7500× *g* for 5 min at 4 °C, and ethanol was carefully removed in each case. Ethanol was then removed by drying the sample at room temperature for 15 min. The sample was dissolved in 50 µL of RNase-free water (Thermo Fisher Scientific, Waltham, MA, USA) and DNA digested by DNase I (Thermo Fisher Scientific, Waltham, MA, USA). The quantity of RNA was determined by measuring absorbance at 260 nm with NanoVue (GE Healthcare, Chicago, IL, USA). In comparison, the quality was estimated by determining the 260/280 nm and 260/230 nm ratios. We only selected samples with both ratios more than 2 for sequencing.

After the QC procedures, ribosomal RNA and long-non-coding libraries were removed using the Illumina Ribo-Zero Plus rRNA Depletion Kit (Illumina, Inc., San Diego, CA, USA). The mRNA was randomly fragmented by the NEBNext^®^ Magnesium RNA Fragmentation Module (New England Biolabs GmbH, Ipswich, MA, USA) followed by cDNA synthesis with SuperScript IV Reverse Transcriptase kit (Thermo Fisher Scientific, Waltham, MA, USA). The second strand was synthesized using NEBNext^®^ Ultra™ II RNA Library Prep Kit for Illumina^®^ (New England Biolabs GmbH, Ipswich, MA, USA). This was followed by purification by AMPure XP beads (New England Biolabs GmbH, Ipswich, MA, USA), terminal repair, polyadenylation, sequencing adapter ligation, size selection, and degradation of second-strand U-contained cDNA by the Thermolabile USER^®^ II Enzyme (New England Biolabs GmbH, Ipswich, MA, USA). The strand-specific cDNA library was generated after the final PCR enrichment and quality assessment. The mRNA library was sequenced with Illumina Miseq (Illumina, Inc. US Illumina) at Novogene Company Limited (Cambridge, UK).

#### 2.5.2. Differential Expression Analysis

The sequencing reads were checked for quality with FastQC [40]. Trimmomatic v0.36 program [41] was used to filter out low-quality sequences (Phred score < 20) and those containing adaptors. The reads were aligned to the *R. phaseoli* strain R650 (NCBI accession: GCA_001664385.1) genome with Bowtie2 v2.4.4 [42]. The alignment file was sorted and indexed by samtools v1.13 [43]). Multcov script in bedtools v2.30.0 was used to count the number of reads overlapping each gene on the reference genome presented in a bed file in all the alignment bam files to generate the count matrix. Contaminating ribosomal RNA sequence counts were removed from the count matrix with an in-house python script before differential expression analysis. DESeq2 v1.30.1 [44] in R v4.1.2 was used to perform count normalization and differential expression. Differentially expressed genes (DEGs) were visualized as a volcano plot using a python package, bioinfokit v1.0.8 [45]. DEGs were treated as significantly expressed when the adjusted *p*-value was less than 0.05 and the fold change was greater than 2.0.

Quantitative real-time PCR (qRT-PCR) was used to check the reliability of the whole RNA-seq process. RNA was extracted with TRIzol^®^ reagent (Invitrogen, Waltham, MA, USA). It was then used to prepare a cDNA library using RevertAid First Strand cDNA Synthesis Kit (Thermo Fisher Scientific, Waltham, MA, USA). The qRT-PCR was performed on the CFX Connect Real-Time PCR Detection System (Bio-Rad, A). Three upregulated DEGs, encoding AMC88_RS25470 (Bon domain-containing protein), AMC88_RS00605 (putrescine-ornithine antiporter, potE), and AMC88_RS16850 (DUF1236 domain-containing protein), were selected for qPCR. The qPCR amplification products of the three genes were normalized to those of gyrase B (*gyrB*) and ATP synthase subunit alpha (*atpA*). One-way ANOVA was performed to determine the significant difference between the expression of Al-treated and untreated controls.

#### 2.5.3. Functional Annotation of DEGs

Protein sequences of DEGs were retrieved from *R. phaseoli* (accession GCF_001664385.1) proteome using the inhouse bash script. These protein sequences were then locally searched against NCBI’s Conserved Domain Database (CDD) [46] with reverse position-specific blast (RPS-BLAST) for conserved domains. These domains were then searched against a database of Clusters of Orthologous Groups of proteins (COGs) [47] for their functional orthology using cdd2cog.pl v0.1 script in Leimbach’s [48] bac-genomics-scripts. The Cello2go [49] and Gram-LocEN [50], web-based systems that screen various amino acid properties on proteins, performed the sub-cellular localization of the proteins. Structure prediction of the first three proteins was implemented with Phyre2 [51], and protein topology was predicted with the Phobius web server [52].

### 2.6. Exopolysaccharide (EPS) Production and Biofilm Formation

In total, 10 mL of four-day cultures grown in defined glutamate-arabinose-galactose-salt media were centrifuged at 10,000× *g* for 10 min at 4 °C. The cells were dried in the oven at 65 °C for 24 h and weighted. Sodium chloride crystals were added to the supernatant up to a concentration of 1 M. An equal amount of isopropanol (10 mL) was added. The solutions were incubated at 4 °C overnight for EPS precipitation. EPS was recovered by centrifugation for 30 min at 15,000× *g* and washed twice in 100% ethanol. After evaporation of all traces of ethanol, EPS was suspended in deionized water. To ensure that all EPS dissolves in water, it was incubated in an orbital shaker maintained at 200 rpm for 3 h at 37 °C. The amount of EPS was then estimated by the phenol-sulphuric acid assay [53,54]. In the tube containing 200 µL of diluted EPS solution, 200 µL of 5% phenol and 98% sulphuric acid were added and thoroughly mixed. The resultant mixture was incubated at room temperature for 1 h, and then the absorption was measured at 490 nm. The quantity of EPS was determined from the glucose standard curve and expressed as the amount in grams of EPS per gram of the cells. For biofilm quantification, 1 mL culture of OD_600_ = 0.3 was put in a test tube followed by treatment with 0 µM, 50 µM, or 100 µM of Al^3+^, and kept in a 30 °C incubator for 48 h. The unbound cells were removed, and the biofilm was washed twice with phosphate buffer (pH 7.0). The biofilm was then stained by 1 mL of gram crystal violet (Sigma Aldrich, Baden-Württemberg, Germany), incubated in the dark for 30 min, and washed three times with the phosphate buffer to remove the unbound stain. It was dried at room temperature for 15 min, and the absorbance was measured at 585 nm with the spectrophotometer.

### 2.7. Intra-Cellular Glutathione (GSH), γ-Aminobutyric Acid (GABA), and Amino Acids Determination

#### 2.7.1. Quantification of GSH

The samples were extracted in 0.5 mL of 5% (*w*/*v*) meta-phosphoric acid. The extract was then diluted in a ratio of 1:10 (*v*:*v*) in water containing ^13^C_2_, ^15^N_1_ labeled GSH as used to analyze GSH in the diluted extracts. Chromatography was done on an Agilent 1200 HPLC system (Agilent Technologies, Böblingen, Germany). Separation of 5 µL of the diluted sample was achieved on a reversed-phase C-18 column (Nucleodur Sphinx RP, 250 × 4.6 mm, 5 µm, Macherey-Nagel, Düren, Germany). Formic acid (0.2%) in water and acetonitrile were employed as mobile phases A and B, respectively. The elution profile was: 0–3.5 min, 2% B in A; 3.5–9.0 min, 2–35% B in A; 9.1–11 min 100% B and 11.1–15 min 2% B in A. The mobile phase flow rate was 1.0 mL/min. The temperature of the column was maintained at 25 °C. The liquid chromatography was coupled to an API5000 tandem mass spectrometer (AB Sciex, Darmstadt, Germany) equipped with a Turbospray ion source operated in negative ionization mode. Infusion experiments with pure standards optimized the instrument parameters. The ionspray voltage was maintained at −4000 eV. The turbo gas temperature was set at 620 °C. Nebulizing gas was set at 60 psi, curtain gas at 30 psi, heating gas at 60 psi, and collision gas at 6 psi. The multiple reaction monitoring (MRM) was used to monitor analyte parent ion → product ion: GSH (*m*/*z* 306.0 →143.0; DP -60, CE -28), ^13^C_2_, ^15^N_1_-GSH (*m*/*z* 309.0 →146.0; DP -60, CE -28). Both the Q1 and Q3 quadrupoles were maintained at unit resolution. Analyst 1.5 software (AB Sciex, Darmstadt, Germany) was used for data acquisition and processing. GSH in the sample was quantified using ^13^C_2_, ^15^N_1_-GSH.

#### 2.7.2. Quantification of GABA and Amino Acids

The samples were extracted in 0.5 mL of methanol. The obtained raw extract was diluted in a ratio of 1:10 (*v*:*v*) in water containing the U-^13^C, ^15^N labeled amino acid mix (algal amino acids ^13^C, ^15^N, Isotec, Miamisburg, OH, USA) at a concentration of 10 µg of the mix per mL. LC-MS/MS directly analyzed GABA in the diluted extracts, according to Scholz et al. [55]. Chromatography was done on an Agilent 1260 HPLC system (Agilent Technologies, Böblingen, Germany). Separation of 1 µL of the diluted sample was achieved on a Zorbax Eclipse XDB-C18 column (50 × 4.6 mm, 1.8 µm, Agilent Technologies, Germany). 0.05% of formic acid in water and acetonitrile were utilized as mobile phases A and B, respectively. The elution profile was: 0–1 min, 3% B in A; 1–2.7 min, 3–100% B in A; 2.7–3 min 100% B, and 3.1–6 min 3% B in A. The mobile phase flow rate was put at 1.1 mL/min. The column temperature was maintained at 25 °C. The liquid chromatography was coupled to a QTRAP6500 tandem mass spectrometer (AB Sciex, Darmstadt, Germany) equipped with a Turbospray ion source operated in positive ionization mode. Infusion experiments with pure standards optimized the instrument parameters. The ion spray voltage was kept at 5500 eV. The turbo gas temperature was set at 620 °C. Nebulizing gas was set at 70 psi, curtain gas at 40 psi, heating gas at 70 psi, and collision gas at medium. Multiple reaction monitoring (MRM) was then used to monitor analyte parent ion → product ion: GABA (*m*/*z* 104.1 →87.1; DP 51, CE 17), U-^13^C,^15^N-Ala (*m*/*z* 94.1 →47.1; DP 51, CE 17). For MRMs of other amino acids: see Appendix A. Both the Q1 and Q3 quadrupoles were maintained at unit resolution. Analyst 1.5 software (AB Sciex, Darmstadt, Germany) was used for data acquisition and processing. Each amino acid’s concentration was calculated relative to its corresponding labelled amino acids. GABA in the sample was quantified using U-^13^C, ^15^N-Ala, applying a response factor of 1.0.

### 2.8. Data Analysis

One-way analysis of variance was performed using Scipy v1.0 [33], followed by Tukey’s HSD for pairwise mean comparisons with statsmodels v0.13.1 [34] in python 3.10. Graphical data representation was performed with seaborn 0.9.0 [32] or Matplotlib v3.5.1 [56] packages in python 3.10. Image adjustment was performed in Gimp v2.10.28 software.

## 3. Results

### 3.1. Rhizobia Exposure to Al Toxicity

Growth in Al-containing media was compared for three rhizobia strains from Western Kenya (S3 and two previously isolated *R. phaseoli strains* B3 and S2) [31] and the commercial strain CIAT899. Al inhibited the growth of all four strains already at 25 µM (~10%), and above 125 µM Al, growth of the four bacteria was inhibited by >90% (Figure 1). Between these two extremes, the three isolates B3, S2, and S3 grew better than the commercial strain CIAT899, and in all Al concentrations, B3 showed the best growth among the three Kenyan isolates. In particular, in media with 100 µM Al, B3 grew almost twice as well as the commercial strain CIAT899, although the percentage growth was not significantly different. Therefore, this strain was investigated in more detail. Figure 1 demonstrates that the new strains are better adapted to the conditions of the Kenyan soil than the commercial strain CIAT899. 

### 3.2. Cell Viability after Al Exposure and Longtime Effect

Next, we compared cell viability and recovery from Al stress for the strain B3 and CIAT899. Short-time exposure (12 h) to all tested Al concentrations had a profound negative effect on the viability of CIAT899, whereas the viability of isolate B3 was not significantly affected by 25 µM Al. However, at 100 µM Al, the viability of B3 was also severely reduced (Figure 2A). Furthermore, recovery from longtime exposure (96 h) to Al differed for both strains. After exposing the cells to 25, 50, 100, or 200 µM of Al, about 87%, 52%, 11%, and less than 1% of the B3 cells recovered, whereas no recovery at all was observed for the standard *R. tropici* CIAT899 (less than 1% recovery at the lowest Al concentration) (Figure 2B). This demonstrates that B3 is better adapted to Al stress than the commercial strain CIAT899. 

### 3.3. Al Binds to the Rhizobium Cell Membrane

Calcein exclusively binds to Al in the pH range between 5 and 6 to generate a fluorescence complex that can be visualized by fluorescence microscopy or quantified with a fluorescence plate reader. Unlike calcein-AM, Calcein does not enter the living cells; thus, it can label extracellular Al bound to the cell membrane. The fluorescence signal produced by CIAT899 was significantly higher than that of B3 (Figure 3). This indicates that more Al is bound to CIAT889 than B3 membranes. 

### 3.4. Membrane Leakage under Al Stress

To investigate whether Al impairs the integrity of the plasma membrane, we determined the amount of extracellular ATP released into the growth medium after exposure of the B3 and CIAT899 cells to Al stress. Isolate B3 released 60 times and CIAT889 350 times more ATP into the medium when the cells were exposed to Al stress (Figure 4A). This suggests that B3 membranes are more stable under Al stress conditions than CIAT899 membranes. 

To test whether protein leakage from the cell increases with increasing Al concentrations in the medium, we used a B3 strain transformed with the fluorescence protein mScarlet-1. The fluorescence signal in the media increased with increasing Al in the media (Figure 4B). These two experiments support the idea that Al makes the membrane of rhizobia more permeable, which results in the release of cytoplasmic substances into the medium.

### 3.5. DEGs in Isolate B3 under Al Stress

When isolate B3 was exposed to 100 µM of potassium Al sulfate (KAl(SO_4_)_2_·12H_2_O) at pH 5.5 for 48 h, 138 genes were upregulated (adjusted *p* ≤ 0.05; log_2_ fold change ≥ 1) compared to the untreated control. Eighty-three genes originated from the chromosome (NZ_CP013532.1), 22 from the plasmid 1 (NZ_CP013533.1), seven from each of the plasmids 1 and 2 (NZ_CP013534.1 and NZ_CP013535.1), and 19 genes from plasmid 4 (NZ_CP013536.1). Only 53 genes were down-regulated under Al stress, from which 39 are on the chromosome, two on plasmid 1 and 6 on plasmids 2 and 4, respectively. Notably, for three genes, coding for the BON domain-containing protein AMC88_RS25470, the putrescine-ornithine antiporter AMC88_RS00605 and the DUF1236 domain-containing protein AMC88_RS16850, we observed very strong stimulations under Al stress (fold changes ~8; Figure 5A). Quantitative real-time PCR confirmed the results for the three genes, with the same order observed by the RNA sequencing (Figure 5B). This suggests that these genes/proteins have important functions under Al stress (cf. below).

### 3.6. Subcellular Localization of Proteins of the DEGs

More than 50% of the upregulated genes code for membrane-associated proteins or are transported across the inner membrane (inner membrane; 31%, outer membrane; 3%, periplasm; 16% and extracellular; 6%). This suggests that Al damages membranes and that the identified genes code for proteins involved in the repair and the prevention of further damage. Moreover, more than 50% of downregulated genes code for cytoplasmic proteins. This indicates that Al enters the cell, inhibits or damages the transcription and translation machinery, and represses the expression of the genes for proteins with cytoplasmic functions (Figure 6).

### 3.7. Structural and Topological Predictions for Proteins of Three Highly Al-Responsive DEGs

Three genes with the most robust response to Al code for a putrescine-ornithine antiporter (AMC88_RS00605), a BON (bacterial OsmY and nodulation) domain (AMC88_RS25470)- and a DUF1236 domain (AMC88_RS16850)-containing protein. The putrescine-ornithine antiporter contains 12 transmembrane regions and is located in the inner membrane (Appendix A). Neither a function nor a structure has been proposed for the other two proteins. AMC88_RS25470 has a single BON domain which has its name from the 20 kDa *E. coli* OsmY protein located in or at the outer membrane or periplasmic space. OsmY is expressed in response to stress and, in particular, osmotic shock. The protein has been proposed to prevent shrinkage of the cytoplasmic compartment by contacting the phospholipid interfaces surrounding the periplasmic space. OsmY has two BON domains, and each of them could interact with the surfaces of phospholipids in the inner and outer membranes [57]. BonA, another BON-domain containing protein from *Acinetobacter baumannii*, is located in the outer membrane and forms a divisome-localized decamer [58]. Loss of BonA modulates the density of the outer membrane, changes its structure, and links to the peptidoglycan. The BonA decamer can permeate the peptidoglycan layer and form a membrane-spanning complex during cell division [58]. In addition, the *Mycobacterium tuberculosis* OmpATb protein is water-soluble, functions as an outer membrane porin, and contributes to the bacterium’s adaptation to the acidic environment [59]. The oligomeric rings are inserted into the phospholipid membrane, similar to related proteins from the Type III secretion systems [60]. Voltage measurements demonstrate that OmpAtb forms conducting pores in model membranes. Our BON domain-containing protein is predicted to be soluble and non-cytoplasmic, with properties resembling proteins destined for the periplasm (Appendix A). It lacks signal peptide, meaning unknown mechanisms transport it across the inner membrane. Finally, 88% of the amino acid residues of this BON domain-containing protein could be modeled to the outer membrane lipoprotein division and outer membrane stress-associated lipid-binding protein (DolP). The strong activation of the gene by Al and the functional analysis of its BON domain suggest that the rhizobial protein might have stabilizing functions in or at the periplasmic side of the outer membrane under stress. 

The AMC88_RS16850 protein has an uncharacterized DUF1236 domain and is predicted to be located in the inner membrane and periplasm. This protein’s first 22 N-terminal amino acids constitute a signal peptide followed by a 20 amino acids-long transmembrane region. The residual C-terminal rest of the protein is predicted to be non-cytoplasmic and thus likely exposed to the periplasmic space (Appendix A). AMC88_RS16850 shows 42% homology to the Outer membrane protein A (OmpA) found in many gram-negative bacteria, which participates in cellular responses to DNA damage, conjugation, protection against viral entry into the host cell, and ion transport (Table 1) [61]. This suggests that the identified protein might have similar functions. Taken together, the strong up-regulation of the three genes in B3 indicates that they have specific membrane-associated functions under Al stress.

### 3.8. Functional Annotation of DEGs

Functional annotation of the proteins of the DEGs according to the COG pathways uncovered that more genes with predicted protein functions are up- than down-regulated (Figure 7). The four most important pathways were “amino acid transport and metabolism”, “carbohydrate transport and metabolism”, “inorganic ion transport and metabolism”, and “cell membrane biogenesis”. This indicates that the cell membrane, primary metabolism, and metabolite transport are the primary targets of Al stress. Interestingly, proteins categorized as “Chromatin structure and dynamics”, “cell cycle control”, “cell division, chromosome partitioning”, or “translation, ribosomal structure, and biogenesis” are mainly downregulated. This demonstrates that Al stress down-regulates general processes associated with gene expression and DNA alterations leading to cell division. It is also necessary to note that many DEGs code for proteins with no or little defined functions (“general function prediction only”, “function unknown”).

About 10.6% of all the DEGs code for ATP-binding cassette (ABC) transporters. These ATP-containing membrane-bound permeases are involved in either export or import of solutes across the periplasmic space [62]. Most upregulated genes which code for proteins of the category “amino acid transport and metabolism” transport or metabolize branched-chain amino acids, histidine, proline, lysine, glycine, serine, glutamate, ornithine, putrescine, or γ-aminobutyric acid. Examples are the lactoylglutathione lyase, which catalyzes the formation of metalloprotein methylglyoxal from GSH, amino acid oxidases, γ-aminobutyrate permeases involved in exogenous GABA uptake, the ABC-type proline, and glycine betaine systems, arginine/lysine/ornithine decarboxylase, and other amino acid transporters. An important observation is the downregulation of mRNA levels for the protoporphyrin-IX and N-acetyl-L-glutamate biosynthesis pathways that utilize L-glutamate through the enzymes glutamate-1-semialdehyde-2,1-aminotransferase and N-acetyl-glutamate synthase. We also observed the downregulation of the mRNA for 4-aminobutyrate aminotransferase, which catalyzes the formation of succinate semialdehyde and L-glutamate from GABA and 2-oxoglutarate. Apparently, Al-stressed cells try to maintain high GABA levels. Repression of dihydrodipicolinate synthase, an enzyme that catalyzes the second step of lysine synthesis from L-aspartate, could be important to maintain high aspartate levels in the stressed cells.

In the carbohydrate transport and metabolism category, the proteins of the upregulated genes are primarily involved in transporting polysaccharide metabolites or proteins involved in carbohydrate metabolism. This includes proteins involved in EPS biosynthesis, ABC-type polysaccharide transport systems with ATPase activities, a lipopolysaccharide biosynthesis protein, polysaccharide export permeases, and an enzyme involved in extracellular polysaccharide synthesis. Higher levels of the latter protein might point to an effort of the cell to synthesize more polysaccharides and export them to the extracellular matrix. Upregulated genes for the inorganic ion transport and metabolism code primarily for ABC-type proteins involved in nitrate, molybdate, sulphate, or Fe^3+^ transport. Others mediate cation transport, particularly for heavy metals such as Cu^2+^, Co^2+^, Zn^2+^, and Cd^2+^. Genes for a putative silver efflux pump and a flavoprotein involved in K^+^ transport were also upregulated. However, several genes for extracellular iron uptake systems were repressed, such as an ABC-type enterochelin transport and a siderophore export system. Also downregulated were genes for Mg^2+^ and Ni^2+^ transport. Finally, in contrast to the upregulation of genes involved in K^+^ uptake, those involved in Na^+^ uptake were downregulated, such as the NhaP-type Na^+^/H^+^ and Na^+^/phosphate symporters.

Upregulated DEGs belonging to the category of cell membrane biogenesis code for small-conductance mechanosensitive channels which either open to allow the solute flow out of the cell or close to prevent solute flow from the cytoplasm. Two important downregulated genes in this category could participate in preserving L-aspartate and L-glutamate in the cytoplasm. UDP-3-*O*-acyl-*N*-acetylglucosamine deacetylase is involved in the biosynthesis of the above-mentioned lipid A, the periplasmic protein, which links the inner and outer membrane, and a membrane fusion protein. Finally, the lysine biosynthesis from aspartate is down-regulated by repression of mRNA level for the enzyme dihydrodipicolinate synthase, and L-glutamate utilization was blocked through the downregulation of the glutamate racemase gene.

### 3.9. EPS Production and Biofilm Formation Increased with Increasing Al Concentration

EPSs are polymers secreted by bacteria to cope with harsh environmental conditions and contain mainly polysaccharides, DNA, and proteins. They are essential components of the complex biofilm, a matrix of extracellular polymers that enable bacterial attachment to surfaces. Some genes responsible for EPS biosynthesis and transport were differentially expressed in the gene expression study. We, therefore, experimentally determined whether the amount of extracellular and biofilm EPS formed by the B3 isolate correlated with the concentration of Al in the medium. There was a very strong correlation between Al concentration and the amount of extracellular EPS (Pearson’s corr. coeff. = 0.9963) and between the amount of EPS and the quantity of formed biofilm (Pearson’s corr. coeff. = 0.9735) (Figure 8).

### 3.10. GSH, γ-Aminobutyric Acid, and Amino Acid Quantification

Upregulation of the genes for glutamate synthase and aspartate aminotransferase under stress might indicate GABA or GSH biosynthesis and accumulation. Therefore, we performed quantitative real-time PCR for the genes for glutamate decarboxylase 1 (GAD1, involved in GABA biosynthesis) and GSH synthetase (GSS, involved in GSH biosynthesis) to compare the expression patterns of the two processes between Al-treated and untreated controls. *GAD1* was higher expressed in treated cells than in untreated cells. In contrast, *GSS* was significantly downregulated in Al-treated cells compared to the untreated cells (Figure 9A). We further quantified intracellular GSH and GABA levels to determine if this is reflected at the metabolite level. According to the results of qPCR analysis, the GSH level was less in Al-treated cells than in the untreated controls (Figure 9B). At the same time, the amount of intracellular GABA was significantly higher in Al-treated cells compared to the untreated controls (Figure 9C). Due to the upregulation of many genes involved in the transport of amino acids across the cell membrane, we also measured the amounts of the amino acids in the Al-treated and untreated cells. Al-treated cells contained lower amounts of most amino acids. The only exception was arginine, which was slightly elevated in Al-treated cells (Figure 9D).

## 4. Discussion

### 4.1. Rhizobia Toxicity to Al

Although Al is the most abundant element in the earth’s crust, it has no known function in living organisms [63]. No known ligands and chaperons can bind Al, there are no known channels or transporters specifically involved in Al transport in or out of biological cells, and no known pathways specific for Al metabolism or excretion [64]. Al is toxic to rhizobia, and our goal was to isolate novel strains which are better adapted to Al-contaminations in soil. In this study, we investigate one of our rhizobial strains which was isolated from bean nodules growing on Kenyan soil. It remains to be determined whether this isolate is unique or only one of many strains which perform better under Al stress.

In our study, all investigated isolates already experienced about 10% growth inhibition at the lowest concentration of Al (25 µM). Even after a short time of exposure, the growth of the commercial strain CIAT899 was severely impaired and recovered only slowly from the stress after longtime exposure. In contrast, the viability of B3 was not significantly reduced when exposed to 25 µM Al for a short time. It also recovered its viability after more prolonged exposure to 100 µM Al. In other studies, CIAT899 tolerated 500 µM of Al^3+^ at pH 5 [65] and 300 µM Al^3+^ at pH 4.5–6 [25]. However, these experiments were performed on agar plates, which often overestimate Al and acid tolerance in rhizobia. Our previous study observed that CIAT899 could grow on YMA plates with pH 3.8, while growth in liquid media was inhibited by >98% at pH 4.8 [31]. Nevertheless, direct comparison of both strains in this study under identical conditions demonstrated that B3 performs better under Al stress than CIAT899. 

About 30% of sub-Saharan African soils are classified as acidic (pH ≤ 5.5) and therefore are faced with severe problems of Al toxicity [66]. Thus, microorganisms proliferating in such soils are challenged to develop resistance against the hazardous H^+^ ions. Since low pHs and the release of Al ions from insoluble compounds are tightly related phenomena, tolerance mechanisms to Al cannot be uncoupled from acidity in soils [67]. Our previous study [31] showed that isolate B3 performs exceptionally well in the media with pH 4.8. Better tolerance to Al in the medium could thus be a direct consequence of its resistance against low pHs.

We found that Al binds to the rhizobia cell membrane, which probably impairs membrane integrity. The binding of Al to the membranes diminished the negative charges of phospholipids and amino acids in membrane proteins in *Triticum aestivum* L. root tips [68]. It also reduced the Mg^2+^-ATPase activity in *Zea mays* L. microsomal fractions [69]. Al binding to membrane phospholipids altered lipid-protein interaction and modified their transport in human erythrocytes [70]. Since significantly more Al bound to CIAT899 than to B3 membrane, we propose that this is responsible for membrane damage and consequently the higher leakage of ATP from Al-treated CIAT899 than B3 cells. Increased membrane permeability under Al stress might explain the lower amount of intracellular amino acids detectable in Al-treated cells than in untreated controls. We also confirmed a high leakage with increasing Al concentrations in the medium at the protein level, with the heterologous protein mScarlet-1. The gene for this water-soluble fluorescent protein was introduced into B3 on a plasmid and is not required for any function of the rhizobium; therefore, the results with mScarlet-1 provide independent evidence for Al-mediated membrane permeability.

### 4.2. Rhizobia Tolerance to Al Toxicity

The high number of DEGs for membrane-associated proteins might be due to the cell’s attempt to repair the Al-mediated membrane damage, restrict further damage, and prevent the entry of Al into the cells. BON domain-containing proteins are structural proteins with stress-protecting functions in or at the periplasmic membrane [57]. For example, DolP, a lipoprotein with an identical structure to AMC88_RS25470, is crucial to the biogenesis of the outer membrane in *E. coli,* and its loss resulted in increased membrane fluidity [71]. BON domains contact the interphases of phospholipids surrounding the periplasmic space preventing shrinkage of the cytoplasmic compartment [71]. Upon osmotic stress, two proteins with BON domains (YgauU and OsmY) were induced in *E. coli* [72], presumably functioning as a membrane-binding domain. In addition to AMC88_RS25470, three additional proteins with BON domains were up-regulated by Al (AMC88_RS03050 [fold change, 2.1], AMC88_RS11410 [fold change, 4.2], AMC88_RS20165 [fold change, 3.1]) in this study. Further experiments will demonstrate whether BON domains participate in membrane stabilization during Al stress in *R. phaseoli*.

We did not find structural or functional information for AMC88_RS16850, a DUF1236 domain-containing protein that is also strongly upregulated at the mRNA level under Al stress. Its structure was weakly linked to that of the outer membrane protein A (OmpA with 42% coverage). AMC88_RS16850 has an uncharacterized binding site and a transmembrane domain. AMC88_RS16850 is predicted to be localized in the inner cell membrane with its C-terminus in the periplasm. It might be involved in the transport/binding of Al or other Al-complexing substances in the periplasm or participate in the ion’s translocation across the inner membrane. Again, this needs to be experimentally determined in future work.

Metabolic processes involved in transcription and protein synthesis were downregulated during Al stress. Downregulation of genes for cytoplasmic proteins may be due to impairments of cytoplasmic functions caused by elevated intracellular Al concentrations. After Al has entered the rhizobia cells through unknown mechanisms, it could bind to DNA [21], RNA, and other transcription/translation machinery components. Of the 14 ABC transporter genes differentially expressed in response to Al in this study, 13 were up- and only one was down-regulated. The latter gene codes for a substrate-binding protein (PstS) of the phosphate ABC transporter family, which is involved in phosphate transportation into the cell [73]. Since Al complexes with more phosphate ions, downregulation of the phosphate importer gene may have a protective function which also restricts Al entry into the cytoplasm. Experiments in *Saccharomyces cerevisiae* and *Schizosaccharomyces pombe* have shown that ABC transporters are involved in heavy metal resistance [74]. In *Arabidopsis thaliana*, mutants lacking AtPDR8 (an ABC transporter gene) exhibited greater tolerance to cadmium and lead toxicity than the wild-type plants [75]. Three genes homologous to known ABC transporters (GenBank ZP_002212691 and NP_766950) induced by various heavy metal ions were identified in *R. leguminosarum, M. loti,* and *S. meliloti* [76]. Therefore, further investigations of the ABC transporters identified in this study may contribute to the understanding of Al extrusion from Al-stressed rhizobia cells.

The enhanced biosynthesis of hydrophobic amino acids was responsible for the tolerance of *Lactobacillus plantarum* against Cd toxicity [73]. The amino acid metabolites spermine, spermidine, and putrescine, which scavenge Pd, relieved a *Halomonas sp.* strain from metal-induced toxicity [77]. In this study, most upregulated genes code for proteins involved in amino acid transport or catabolism, such as amino acid oxidases, which catalyze the oxidation of L-amino acids to their corresponding α-keto acids, amino acid decarboxylases, which catalyze the formation of indolamines, catecholamines, and trace amines [78], as well as many other genes involved in the transport of amino acids. Transport and metabolism of amino acids might also explain the low amount of intracellular amino acids observed in the Al-treated cells. The exception was amino acid L-glutamate, whose biosynthesis genes were upregulated in Al-treated cells in addition to increased extracellular glutamate uptake.

Additionally, we observed downregulation of most genes that utilize glutamate except those involved in GABA biosynthesis. Elevation of GABA biosynthesis (GAD1) from glutamate and further repression of genes involved in GABA degradation can explain the elevated amount of GABA in Al-treated cells. However, we did not observe an increased GSH biosynthesis from glutamate in Al-treated cells. Expression of the *GS* gene was not downregulated in real-time qPCRs, and significantly less GSH accumulated in Al treated cells than in untreated controls. These data suggest that GSH might not be involved in or at least might be less critical for rhizobial tolerance to Al stress. However, GABA seems to participate in Al tolerance in *R. phaseoli* through unknown mechanisms.

Transport and metabolism of carbohydrates may also play a role in Al tolerance in rhizobia. Notably, higher expression of EPS and lipopolysaccharide biosynthetic genes and genes involved in the extracellular transport of the polysaccharides were observed in Al-treated cells. Several studies had previously established the involvement of EPS in rhizobia tolerance to Al toxicity [22,76]. However, an involvement of EPS in Al tolerance does not seem to be widespread in rhizobia; since several high EPS producers were sensitive to Al, and some Al-tolerant EPS knockdown mutants still tolerated high concentrations of Al [25]. In our study, the amount of EPS production in *R. phaseoli* positively correlated with the concentration of Al in the medium. This, therefore, suggests that rhizobial tolerance is species-specific, and our results cannot be applied to all rhizobia species. Mechanisms through which EPS relieves rhizobia from Al stress are poorly understood. The idea that Al complexes with EPS making it unavailable and thus becoming less toxic was revised after discovering that the addition of EPS to the media failed to relieve Al-sensitive rhizobia from Al toxicity (cited by [25]). An alternative hypothesis could be that the produced EPS promotes biofilm formation, restricting Al penetration into the rhizobia cell. In fact, biofilm formation positively correlated with the Al concentration in the media. EPS protection by the rhizobia may thus be a mechanical block rather than chemical complexation of Al ions.

Several genes involved in the transport of Zn, Cd, Cu, and Co were induced by Al stress. This may show that transporters involved in extruding excess metal ions also participate in removing Al from the cells. In *E. coli*, resistance-nodulation-cell division (RND) superfamily efflux pumps spanning the inner and outer membranes were responsible for *E. coli*’s resistance to several toxic heavy metal ions [26]. Moreover, we observed repression of genes involved in the uptake of different extracellular iron, such as the ABC-type enterochelin transport system and a siderophore export system. Siderophores, usually complex and enabling cellular uptake of extracellular iron, also complexed Al and transported it into *Bacillus megaterium* cells [79].

## 5. Conclusions

We have shown that the rhizobia cell membranes are primary Al targets and responsible for the membrane damage. The cells activate mechanisms that repair and stabilize the membrane. Increased extracellular EPS is used to form a more robust biofilm around the cells, restricting further Al entry. Since Al seems to be transported into the cells using similar mechanisms to iron, both iron and Al uptake are downregulated in Al-stressed cells. We identified ABC-transporters and novel proteins associated with the bacterial inner and outer membranes, which are candidates for conferring Al tolerance to rhizobia. Downregulation of genes for cytoplasmic enzymes involved in primary and amino acid metabolism are consistent with alteration in levels of their target metabolites. Besides general repression of cytoplasmic functions under Al stress, specific metabolites such as GABA might play uncharacterized roles in the Al resistance response in *R. phaseoli*.

## Figures and Tables

**Figure 1 cells-11-00873-f001:**
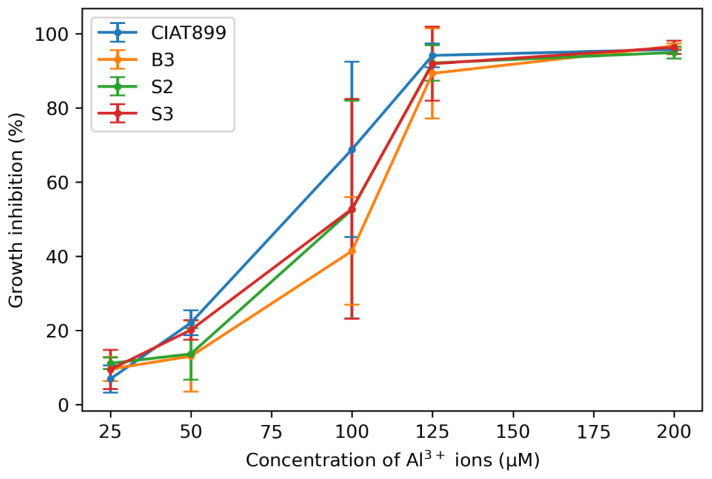
Percent growth inhibition of isolates B3, S2, S3, and standard *R. tropici* CIAT899 in media with Al at pH 5.5.

**Figure 2 cells-11-00873-f002:**
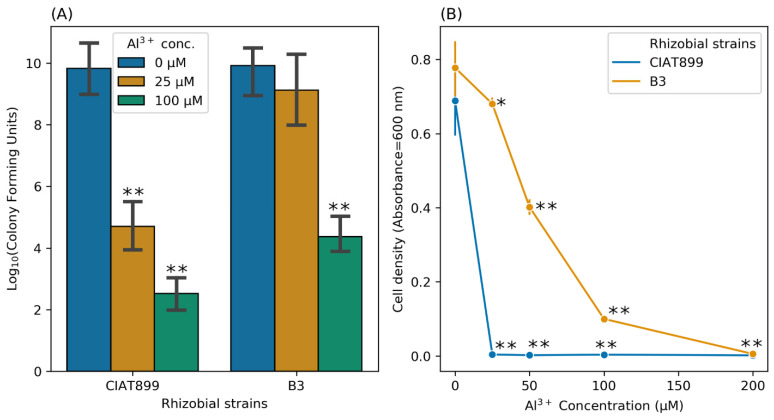
Cell viability approximated by plate count of viable cells after short-time exposure (**A**) and viability recovery after longtime exposure (**B**) of isolates B3 and CIAT899 to different Al concentrations. Asterisks represent significance differences between Al-treated CIAT899 and B3 and their respective untreated controls (at 0 µM Al) (one-way ANOVA; * *p* ≤ 0.05; ** *p* ≤ 0.001).

**Figure 3 cells-11-00873-f003:**
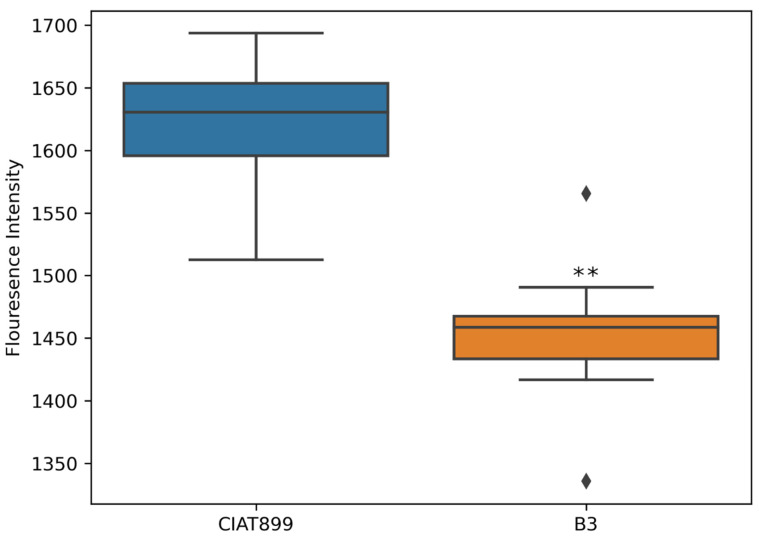
The fluorescence signals from Calcein bound to the plasma membranes of Al treated rhizobia cells and untreated control exemplarily quantified with a fluorescence plate reader. Asterisks represent significance differences between fluorescence intensity from CIAT899 and B3 (one-way ANOVA; ** *p* ≤ 0.001).

**Figure 4 cells-11-00873-f004:**
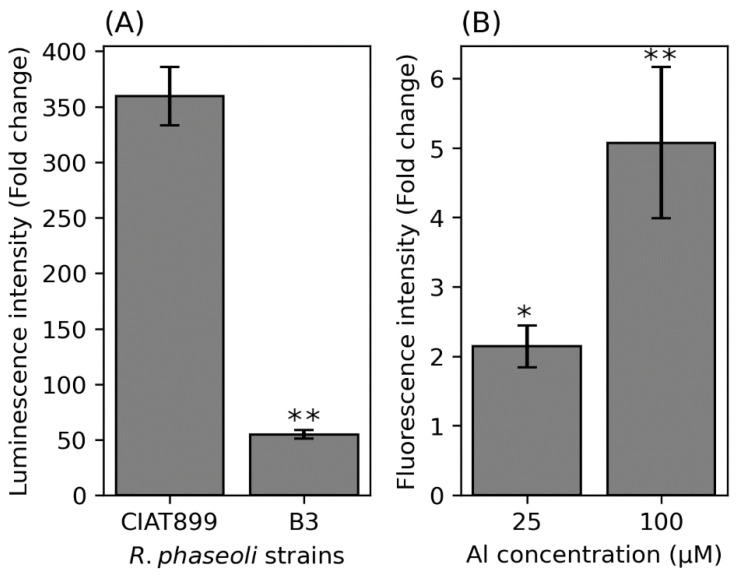
Extracellular ATP (**A**) and extracellular mScarlet-1 (**B**). Asterisks represent significance differences between luminescence intensity from CIAT899 and B3 (**A**) and fluorescence intensity between untreated B3 cells with Al-treated cells at various concentrations (one-way ANOVA; * *p* ≤ 0.05; ** *p* ≤ 0.001). Values on the *y*-axis show fold change compared to the untreated cells.

**Figure 5 cells-11-00873-f005:**
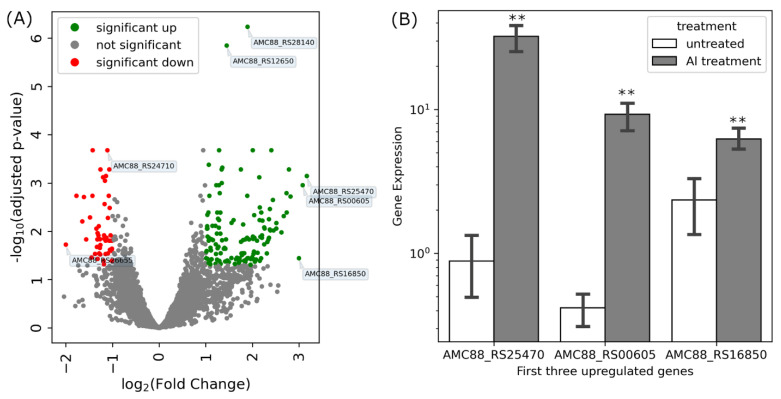
Gene expression analysis of Al-treated B3 cells compared to untreated control cells. A volcano plot (**A**) shows significantly different (colored) or not significantly different (grey) up-and down-regulated genes in Al-treated cells. (**B**) Quantitative real-time PCR of the three upregulated genes, highlighted in panel (**A**). Asterisks in (**B**) represent significant differences in transcript abundance between Al treated samples and untreated controls (one-way ANOVA; ** *p* ≤ 0.001).

**Figure 6 cells-11-00873-f006:**
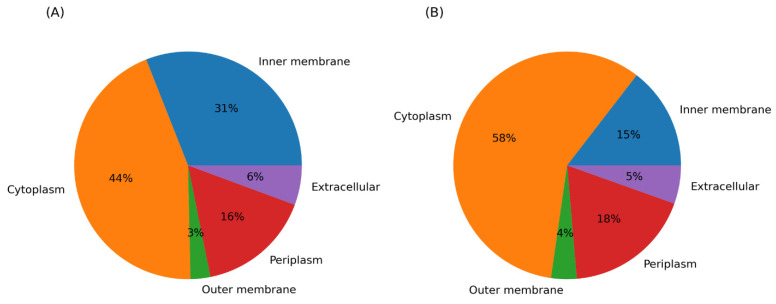
Subcellular protein localization of the proteins of the DEGs, (**A**) proteins of upregulated and (**B**) downregulated genes.

**Figure 7 cells-11-00873-f007:**
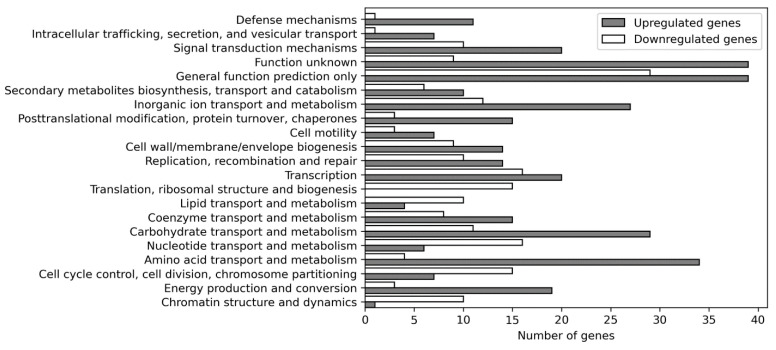
Functional annotation of proteins of DEGs by the cluster of orthologous groups.

**Figure 8 cells-11-00873-f008:**
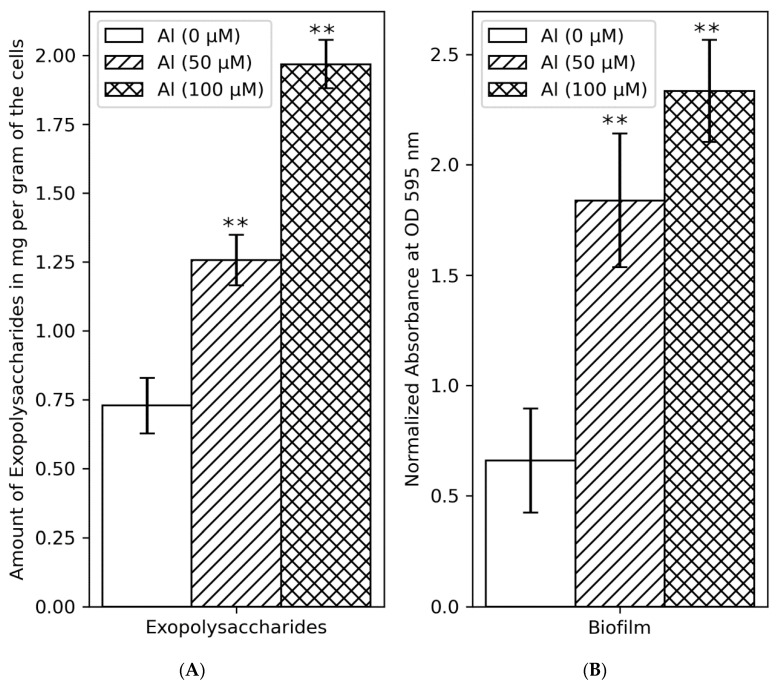
Amount of produced exopolysaccharides (**A**) and biofilm formation (**B**) by isolate B3 under different concentrations of Al. Asterisks represent the significant differences in the quantity of EPS and biofilm between Al-treated samples and untreated controls (one-way ANOVA; ** *p* ≤ 0.001).

**Figure 9 cells-11-00873-f009:**
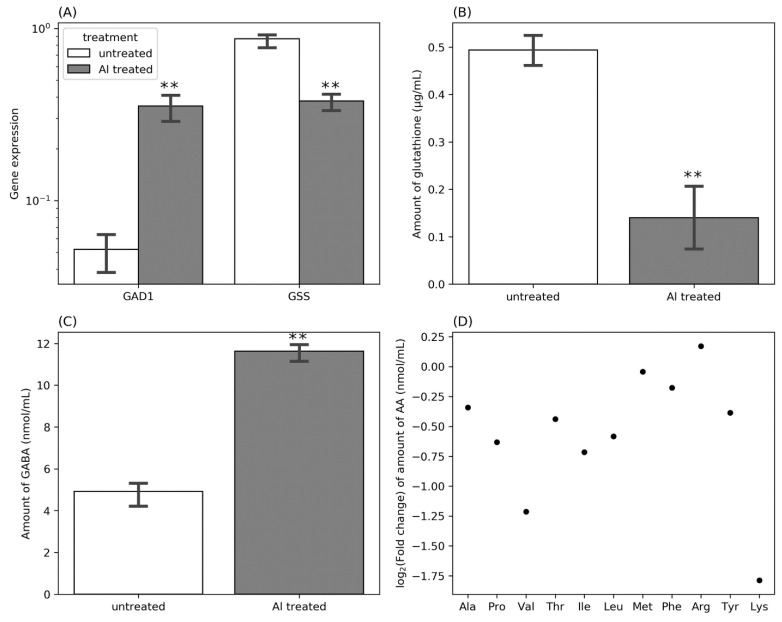
Expression of *GAD1* and *GSS* genes by RT-qPCR (**A**), fold change of the amounts of GSH (**B**), γ-aminobutyric acid (**C**), and amino acids (**D**) in Al-treated cells relative to the untreated controls. Asterisks represent significant differences in the expression levels of *GAD1* and *GSS* genes (**A**), amount of GSH (**B**), and GABA (**C**) between Al-treated samples and untreated controls (one-way ANOVA; ** *p* ≤ 0.001).

**Table 1 cells-11-00873-t001:** Structural prediction of the three proteins whose mRNAs responded most strongly to Al stress. AMC88_RS25470 (BON domain-containing protein), AMC88_RS00605 (putrescene-ornithine antiporter protein), and AMC88_RS16850 (DUF1236 domain-containing protein).

Structural Modeling		AMC88_RS25470	AMC88_RS00605	AMC88_RS16850
Signal peptide		×	×	√
Transmembrane		×	√	√
Cytoplasmic regions		×	√	×
Non-cytoplasmic regions		√	√	√
Subcellular localization		periplasm	cytoplasm, inner membrane, and periplasm	the inner membrane, periplasm
Predicted protein function		binding (phospholipids?)	transmembrane transport	binding (arginine, metal ions?)
Structural homolog	Model template	DolP	arginine/agmatine antiporter	OmpA
	Coverage (%)	88	91	42

## Data Availability

The data presented in this study are openly available in the National Center for Biotechnology Information (NCBI) Gene Expression Omnibus (GEO) repository (GEO accession number: GSE193556).

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
