# Peer review of "The Cell Membrane of a Novel Rhizobium phaseoli Strain Is the Crucial Target for Aluminium Toxicity and Tolerance"

_cells, 2022, doi:10.3390/cells11050873_

Round 1

Reviewer 1 Report

"Expression profiles showed that the
primary targets of Al are genes involved in membrane biogenesis, metal ions binding and transport,
carbohydrate, and amino acid metabolism and transport." - this covers almost everything. This is because there is no sense to compare the transcriptome of two different strains directly. 

"Guided by differential gene expression, we assayed the intracellular γ–aminobutyric acid (GABA), glutathione (GSH), and amino acid levels, determined the amount of extracellular exopolysaccharide (EPS) and biofilm formation during Al stress." - Not clear, the preceding statement says everything was differentially expressed, then how it guided to GABA and GSH selection.

"In conclusion, we demonstrate that Al tempers with the membrane integrity of rhizobia and, therefore, metabolic processes targeting membrane protection, repair, and
stability are essential for Al tolerance in R. phaseoli." - I don't think there is any novelty in this conclusion. 

The authors need to elaborate further to clarify how frequent is the Al resistant rhizobium. From how many screened rhizobia they got the B3 strain 

What is the possibility for natural mutation in B3, S2, S3, and CIAT899 while growing on Al media? 

Please add denotations for significant differences. Looking at a large standard deviation, I don't think there is any statistically significant difference among these strains at different concentrations.

Figure 2 - better to use different letters to indicate significant difference, see in Fig2 b at 200 concentration it is not clear the ** is for strains or concentration. 

The authors need to improve Figure 3, There should be a considerable number of cells to quantify the fluorescence.  

Reviewer 2 Report

The manuscript “The Cell Membrane of a novel Rhizobium phaseoli strain is the crucial target for aluminium toxicity and tolerance” by Wekesa et al sent for publication to Cells deals with important topic like Al toxicity, soils and important crop such as common bean, and nitrogen fixation and will be of interest to the scientific community working on the topic.

The manuscript deserve to be published after minor revisions. Below are my remarks.

In L 211, the authors wrote that they used fold change greater than 2.0 in DEG analysis but L338, they wrote  that 138 genes were upregulated with fold change >2. This need some clarification.

Introduction:

The introduction is well written and point the main problems related to the topic.

Materials and methods

M&Ms are somehow well written.

Results:

Results are clearly presented.

Discussion

This part is well written.

Typos:

L112: Instead of “1 ml” please use One ml

L186: Instead “DNAse 1” it is better DNA se I.

Overall, I would like to congratulate the authors for the good job that they performed.

Author Response

Point 1: In L 211, the authors wrote that they used fold change greater than 2.0 in DEG analysis but L338, they wrote  that 138 genes were upregulated with fold change >2. This need some clarification.

Response 1: In L211, we used the term fold change greater than 2 while in L338, we used the term Log2 Fold change. The value that results when Log2 Fold change is evaluated is equal to 2, they are therefore technically equal and have the same meaning

Point 2: L112: Instead of “1 ml” please use One ml

Response 1: Corrected in the manuscript

Point 3: L186: Instead “DNAse 1” it is better DNase I

Response 3: Corrected in the manuscript

Round 2

Reviewer 1 Report

The authors have addressed all of my concerns the revised version of the MS looks appropriate. I recommend acceptance of the revised version of MS.